# Six Main Contributing Factors to High Levels of Mycotoxin Contamination in African Foods

**DOI:** 10.3390/toxins14050318

**Published:** 2022-04-29

**Authors:** Queenta Ngum Nji, Olubukola Oluranti Babalola, Theodora Ijeoma Ekwomadu, Nancy Nleya, Mulunda Mwanza

**Affiliations:** 1Food Security and Safety Focus Area, Faculty of Natural and Agricultural Sciences, North-West University, Private Bag X2046, Mmabatho 2735, South Africa; queenbrighten@yahoo.com (Q.N.N.); 23115394@nwu.ac.za (T.I.E.); 27545466@nwu.ac.za (N.N.); mulunda.mwanza@nwu.ac.za (M.M.); 2Department of Animal Health, Faculty of Natural and Agricultural Sciences, North-West University, Private Bag X2046, Mmabatho 2735, South Africa

**Keywords:** mycotoxins, African foods, climate change, awareness, farming systems, regulatory limits, detection techniques, storage practices

## Abstract

Africa is one of the regions with high mycotoxin contamination of foods and continues to record high incidences of liver cancers globally. The agricultural sector of most African countries depends largely on climate variables for crop production. Production of mycotoxins is climate-sensitive. Most stakeholders in the food production chain in Africa are not aware of the health and economic effects of consuming contaminated foods. The aim of this review is to evaluate the main factors and their degree of contribution to the high levels of mycotoxins in African foods. Thus, knowledge of the contributions of different factors responsible for high levels of these toxins will be a good starting point for the effective mitigation of mycotoxins in Africa. Google Scholar was used to conduct a systemic search. Six factors were found to be linked to high levels of mycotoxins in African foods, in varying degrees. Climate change remains the main driving factor in the production of mycotoxins. The other factors are partly man-made and can be manipulated to become a more profitable or less climate-sensitive response. Awareness of the existence of these mycotoxins and their economic as well as health consequences remains paramount. The degree of management of these factors regarding mycotoxins varies from one region of the world to another.

## 1. Introduction

Mycotoxins are toxic secondary metabolites produced by certain species of fungi and continue to attract global attention because of their significant economic importance, impact on human health, animal productivity, and domestic and international trade [1]. The most common mycotoxins in African foods include the following: aflatoxins; fumonisins; patulin; ochratoxin A; deoxynivalenol; and zearalenone, among others. Cereals are the major food susceptible to mycotoxin contamination, with maize being the main cereal [2]. About 25% of food crops have been reported by FAO (2004b) to be contaminated with mycotoxins. The validity of this figure (25%) is a matter of scientific discourse. This figure has long been cited as far back as 1985 [3]. Eskola et al. [4] conducted a study on the origin and rationale of the 25% and found no accurate published data on the estimate or information on which the dataset was based or how it was calculated. Earlier on, [5] estimated 40% productivity loss in developing countries due to diseases exacerbated by moulds, but recent estimates are higher, with Eskola et al. [4] reporting losses between 62% and 80%. Although economic losses associated with mycotoxins are difficult to ascertain, especially in Africa with complex food and agricultural practices, some researchers have indicated extremely high losses worth billions of dollars [1,6]. For instance, years ago, in the United States of America, annual losses resulting from all mycotoxin-related issues were estimated to be as high as USD 1.4 billion [7] and recently Human [8] estimated USD 1 billion annually on feed alone. Senerwa, Mtimet [9] estimated loss of USD 17.28 million due to mycotoxin contamination in the Kenyan dairy industry. 

Africa’s geographic location in the tropics, combined with harsh and changing climatic conditions in this region, exposes agricultural products to mould contamination, resulting in the production of enormous amounts of these toxic chemicals by the fungi for their survival. Production of mycotoxins is not necessary for fungal growth but is considered a defence mechanism against predators and changes in the ecological niche of the fungi [4]. Thus, climate is important for fungal colonisation of agricultural commodities and production of mycotoxins. In addition to climate change, other environmental and human factors, such as frequent occurrence of droughts, pest infestations, improper agricultural and storage practices, socioeconomic status, low-level of awareness, detection techniques, lack of regulatory bodies and enforcement mechanisms, chronic food insecurity, and political and economic instability, among others, have hindered efforts to reduce fungal infestation and mycotoxin contamination in food crops and products. 

Though a lot of work has been done on mitigation methods of mycotoxins in food in Africa, stakeholders (farmers, traders, processors, and consumers) acquire and assimilate knowledge differently; thus, appropriate practices are not sufficiently implemented as intended [10,11]. While there could be a number of factors contributing to the low knowledge levels and poor implementation of the preventive and control measures of mycotoxins among various stakeholders, some authors have clearly stated that regulatory guidelines are too complex to be understood by non-scientific and most audiences in the region [10,12]. In addition, most of the mitigation methods are imported from developed countries where studies were carried out for that specific geographical location as per their climatic condition, type of mycotoxins, and consumption habits according to that location. Hence, the mitigation methods applied in developed countries cannot realistically be implemented in Africa because of the harsh climate, nature of the farming and or food systems, socio-political issues, poverty, and civil wars, among others. Therefore, re-evaluation of mitigation strategies need to take into consideration the African reality, such as sustainability, cultural acceptability, and economic feasibility, as well as ethical implications, since this is what defines and gives value to an African. 

Africa is one of the regions highly affected by mycotoxins contamination; thus, the purpose of this review was to explore major factors that contribute to high incidence of mycotoxin contamination in food in Africa—factors such as climatic and environmental factors, farming systems, pre-harvest and post-harvest handling techniques, storage techniques, regulatory limits, detection techniques, and socio-political factors, as well as awareness. Some of these factors are intertwined; for example, farming systems or techniques (rain-fed or irrigation) are determined mainly by climatic factors, such as rainfall and temperature. These decisions are influenced by cultural beliefs based on subsistence farming. Awareness of the effects of mycotoxin contamination of food in a country depends, to some extent, on the socio-political nature of that country. Thus, this review seeks to explore factors that contribute the most to the high prevalence of toxins in food in Africa.

## 2. Factors Responsible for High Levels of Mycotoxin Contamination in African Foods

Africa continues to record high incidences of liver cancers globally, which is directly linked to consumption of mycotoxin-contaminated foods [13]. Production of mycotoxins is climate-sensitive and Africa has high vulnerability to the impact of climate change because it depends solely on weather and climate variables for agricultural production. Knowing the different factors responsible for high levels of these toxins in African foods will be a good starting point for effective mitigation of mycotoxins in Africa. While there could be a number of factors that contribute to high incidences of mycotoxin in African foods, six factors were identified as being responsible for these high incidences as follows: climatic and environmental factors; farming systems/processing and storage techniques; mycotoxin detection techniques; mycotoxin regulatory limits; socio-political factors; and awareness.

### 2.1. Climatic and Environmental Factors

Authors have reported with certainty that climate change is the main agro-ecological driving force of fungal colonisation and mycotoxin production [14,15]. Fungi colonise many crops and are adapted to a wide range of environmental conditions, with specific as well as overlapping ecological niches [16]. The extent of fungal growth and aflatoxin production in cereals depend on temperature, moisture, soil type, and storage conditions [14]. High temperatures, greater carbon dioxide (CO_2_) concentrations, drought stress, and rainfall directly affect maize and the prevalence of *Aspergillus flavus*, favouring fungal growth, conidiation, and spore dispersal and impairing the growth and development of maize [17,18]. Frequent and extended periods of drought stimulate production of mycotoxins under both pre- and post-harvest conditions [19,20,21]. Overall, the impact of climate change, for example, on maize output is influenced by factors such as rainfall, pests, diseases, and temperature [22]. Zuma-Netshiukhwi, Hlazo [23] state that an increase in temperature by 1 °C or 2 °C leads to approximately 20–25% reduction in grain yield, while other scholars estimate up to 50% reduction in yield, depending on the reproductive stage of the plant. [24,25]. When intense reduction in precipitation is accompanied by higher temperatures, it results in more frequent and intense droughts [26]. Rainfall, for instance, was projected to decline by approximately 15% by the end of the 21st century [27]. Boko [28] envisaged that Africa will become 5% to 8% more arid and semi-arid, which perhaps, will cause an increase in drought and may lead to increased crop stress and mycotoxin contamination [19,29]. Furthermore, available climate data indicate that atmospheric concentrations of CO_2_ are expected to double or triple (from 350 to 400 ppb to 800 to 1200 ppb) in the subsequent 25–50 years. Different regions are anticipated to face increases in temperature, coupled with elevated CO_2_ (800–1200 ppm) and drought episodes, with concomitant effects on pests and diseases as well as crop yield [30]. Additionally, the atmosphere and soil moisture are affected by evapotranspiration changes due to expected temperature and precipitation changes. These changes lead to a higher moisture content of the atmosphere at a rate of about 7% for every 1 °C rise [31], which favours the growth of fungi and production of mycotoxins. The average soil moisture is expected to decrease annually in the Mediterranean region and subtropics and is expected to increase in east Africa, central Asia, and some other regions with increased precipitation [32]. The literature shows that mycotoxin contamination in cereal grains will increase with climate change differently at regional levels and the climate scenario considered [33,34]. Changes might differ by scenario and region, but generic changes include increase in temperatures between 1.1 and 6.4 °C by the end of the century [32,35].

Unfortunately, sub-Saharan Africa has been reported as a region of higher vulnerability to the impact of global climate change because of its total dependence on weather and climate variables for agricultural production [36]. In Zambia, for example, higher mycotoxin concentration in groundnut was primarily a consequence of climatic extremes, such as serious drought and hot summer temperatures [19]. Moreover, in Kenya, it was found that maize and sorghum grown in semiarid tropical regions were more liable to mycotoxin contamination than in temperate regions [37]. Thus, in already hot climates, more frequent drought will result in higher production of mycotoxins [21]. In Malawi, Matumba, Sulyok [38] found that maize harvested from regions with high temperatures and low rainfall were highly contaminated with mycotoxins. For example, north and southern Africa have a drier climate, and the incidence of mycotoxins exposure is lower compared with the levels observed in east and west Africa, which have high temperatures and humid climate [39]. In addition, in Serbia, high mycotoxins in maize were as a result of high temperatures and occurrence of drought [40]. South Africa is a water-stressed country, with high spatio-temporal rainfall variability and is classified as a predominantly semi-arid country [41]; temperatures are projected to rise at 1.5 to 2 times the global rate [42]. Additionally, there is frequent occurrence of droughts in the country [43]. For instance, in the summers of 2010, 2011, 2012, 2015, and 2017, South Africa suffered severe droughts [42,43,44] that led to low crop yields. Unfortunately, analyses of mycotoxins are absent for most of these years, but in the years 2010 and 2015, there was occurrence of drought, and maize samples cultivated in these years were analysed and samples were reported to be contaminated with aflatoxins [45,46]. Furthermore, Meyer et al. (2019) had performed a long-term analysis of mycotoxins and reported aflatoxin contamination only for the year 2015. Independent studies by Meyer, Skhosana [47] revealed that other samples from non-drought years were not contaminated with aflatoxins except those of the year 2015. Studies elsewhere showed that increases in levels of aflatoxins correlated with decreases in rainfall and increases in ambient temperature [19,48,49]. As in Table 1, the year 2015, for example, experienced a decrease in rainfall and an increase in average temperature [50]; there was severe occurrence of drought as well. Hence, the aflatoxin contamination of maize could be attributed to the consequences of climate change.

With gradual increases in temperatures, with aridity predicted to increase between 5% and 15% [27,28] and decrease in rainfall in the continent, which are enabling conditions for fungal growth, it is obvious that climate change is the primary reason for high levels of mycotoxins in African foods.

### 2.2. Farming Systems, Processing, and Storage Techniques

#### 2.2.1. Farming Systems

Farming systems entail an integrated strategic detail of resource management of crops, trees, and animals, along with labour and capital to optimise the use of land resources [51]. Farming systems are defined based on: available natural resources (including water, land area, soils, elevation, and length of growing period); population; cropping and pasture extent; the dominant pattern of farm activities and household livelihoods; and access to markets. In line with the scope of this study, the farming systems, though numerous, are classified primarily based on water requirements: irrigation and non-irrigation or rain-fed. Given the arid and semiarid nature of most of sub-Saharan Africa, water is always of utmost importance in the production of the region’s agricultural output, considering the fact that global warming is an on-going event. The most affected farming systems are likely to be those in arid, semiarid, and dry sub humid areas [52].

##### Non-Irrigation Farming

Non-irrigated farming is not supplied with water by artificial means but rather depends on rainfall for crop cultivation. More than 90% of Africans practice non-irrigation farming, which is susceptible to climate change. Smallholder or subsistence farmers who depend on rain-fed agriculture are already experiencing the effects of climate change [53]. This could be one of the reasons of high levels of mycotoxin contamination in African foods. With regard to mycotoxin contamination in food, there are seldom sharp boundaries between climate change and farming systems since meteorological conditions remain the most important factor in the production of mycotoxins and rainfall is one of them. Serious climatic variations in most African countries, classified as semi-arid to arid (as shown in previous studies), revealed an increase in aridity between 5% and 15% [27,28]. Thus, drastic rainfall variations as temperatures increase may cause an increase in droughts and definitely have an impact on water available to the growing crop, leading to increased crop stress and possibly mycotoxin contamination. In non-irrigated farming, the chances of stress to the plant, if the rainfall is low, are high. This can lead to plants that are easily infected by moulds, with subsequent mycotoxin production if not substituted with irrigation. Cultivation of maize, for example, in many parts of Africa, is common during the rainy season, as necessity in rain-fed systems can lead to an increase in accumulation of aflatoxins [54]. Additionally, high temperatures and drought stress have directly affected maize and the occurrence of *Aspergillus flavus,* favouring fungal growth, conidiation, and spore dispersal and impairing the growth and development of maize [17,19,20,21]. Thus, in Africa, with the current climatic scenario, non-irrigated farming could be the most affected since water reduction not only will affect crop production yield but also will encourage the proliferation of fungal and production of mycotoxins.

##### Irrigation Farming

Irrigation is the agricultural process of artificially applying controlled amounts of water to land to assist in the production of crops, giving them a massive boost compared with crops in the rain-fed system. Proper and uniform irrigation of crops reduces drought stress. Studies have shown that crops grown under drought-stressed conditions have higher mycotoxin concentration [19,20]. Even though irrigation in Africa has the potential to boost agricultural productivities by at least 50%, food production in the region is almost entirely rain-fed, with irrigation areas taking just 6% of the total cultivated area. Irrigation schemes are expensive to acquire by smallholder African farmers. In 2007, the World Bank examined detailed cost and performance data from several investment banks for 314 irrigation schemes in 50 countries in Asia, Africa, and Latin America [55] and confirmed that overall unit costs were significantly higher in sub-Saharan Africa than other regions. Even though, the potential for irrigation in Africa is highly dependent on factors such as geographic, hydrologic, agronomic, and economic factors, Kikuchi, Mano [56] evaluated the economic viability of constructing large-scale irrigation schemes in Africa and concluded that at current and likely future prices of crops, irrigation schemes are not economically viable investments. Sustained quality operation and maintenance requires adequate finance as well as human resources, which is in short supply in sub-Saharan Africa. In Africa, a region classified as arid to semi-arid, having only 6% of total cultivated irrigation areas might just be another reason for the high contamination of African foods by mycotoxins. This can also be explained by the environmental stress they succumb to during periods of growth. Under normal circumstances, during periods of droughts or in water scarce regions, agriculture often relies on irrigation for its water requirements. The authors of [57] ranked irrigation and water management as critical factors in planting maize. Studies have shown that levels of mycotoxin in maize could be reduced by 99% when a combination of appropriate irrigation and insecticide is applied, compared with non-irrigated, non-treated maize. Thus, to optimise irrigation cost and to reduce plant stress and risk of mycotoxin contamination of crops, supplemental irrigation is highly recommended [58], especially during the grain filling period, as it is crucial for agronomic practices to lessen the severity of drought or exposure of the crop to high temperatures, in order to minimise the risk of mycotoxin contamination [48]. 

#### 2.2.2. Processing Techniques

##### Pre-Harvest

There are no conventional techniques for preventing the formation of mycotoxins during pre-harvest stages of crop production, but several non-conventional mycotoxin risk pre-harvest preventive techniques and good agricultural practices have proved to be effective. Pre-harvest techniques include the following: crop rotation; intercropping; residual management; sowing time; early harvest; appropriate use of fertilizers; insecticide and herbicides, among others. Crop rotation and intercropping minimise mycotoxin contamination by breaking the infectious cycle; hence, knowing the right crop combination can go a long way in managing mycotoxin contamination. Rotation of legume crops (cowpea and soybean) with maize can help break pest and disease cycles and improve soil fertility [57,59]. According to Mutiga, Were [60], intercropping of cowpea, beans, and maize lowers aflatoxins in maize compared with maize grown as a sole crop, since the amount of nitrogen in the soil is improved. Residue or debris from previous harvests contain fungal spores, which remain dormant in the soil between crops from year to year [61]. Thus, proper management of previous harvest residue by either removing or burying will help minimise fungal infection in the field [62,63]. Furthermore, adjusting sowing time or selecting an appropriate cultivar for lower temperature and water stress conditions will help reduce the effects of mycotoxin contamination of crops, as well as helping to avoid wet periods during early flowering, if possible. Early harvesting decreases fungal infection and mycotoxin formation in crops in the field and is highly recommended in high-risk regions, as fungal pathogens will have less time to develop and potentially produce aflatoxins [21,64]. For example, a report by Negash [65] revealed that aflatoxins in maize increased 4- to 7-fold after 3–4 weeks delay in harvest after maturity. Agricultural practices such as overcrowding of plants should be avoided, as it may lead to humid and warm conditions, which favour insects, fungi, pathogens, and diseases, resulting in reduced yields due to competition for soil nutrients and sunlight, especially during drought stress [59,62]. Appropriate application of fertilizers and soil additions, such as lime, animal manure, and compost, can minimise plant stress, especially during seed development, by assuring an adequate soil pH and plant nutrition. Insects are capable of carrying spores of mycotoxin-producing fungi from one plant to another; therefore, appropriate control of insect pests would reduce levels of mycotoxin contamination [66].

##### Post-Harvest

Overall, post-harvest losses in sub-Saharan Africa are estimated to be between 40% and 80% [67,68,69]. Mycotoxin post-harvest control stages are a challenge, and this is one of the main reasons why mycotoxin occurrence is high in Africa, due to lack of infrastructure for adequate drying and proper storage. Thus, reducing post-harvest losses can be a solution to achieving food security and safety with regard to mycotoxins contamination in Africa. Maize, for example, is one of the major staple foods in Africa that is affected by high levels of post-harvest losses and mycotoxin contamination. Maize is usually harvested with high moisture content, ranging between 19% and 25% [70], and provides an ideal condition for grain germination, insect infestation and multiplication, and growth of moulds; but then, a moisture level of 13% or below, is required for storage [71]. Harvesting should be as quick as possible, especially during the rainy season, and care must be taken to prevent damage during harvesting, as damage to the cobs permits easy entrance of fungi [72]. Maize should be harvested manually without removing the husk in order to reduce aflatoxin contamination [65,73]. Most fungal attack occurs during harvest because of dropping and drying cobs on bare ground or cobs left to dry in the field. This allows for easy transfer of the fungus from the soil to the storage facilities [74,75]. Field stacking (heaping) does not provide enough aeration for the ears and could lead to colonisation of maize by aflatoxigenic fungal strains [71]. Maize ears should not be left in containers for more than 6 h between harvesting and drying [72]. 

Drying has been ranked as the most important postharvest management action by Logrieco, Battilani [57]. Unfortunately, in Africa, substandard or crude drying methods—such as open-air drying, hanging maize cobs, use of sheeting, and on-field drying—are still being practised and might have contributed a great deal to the high percentages of post-harvest losses in addition to the high incidence of mycotoxins contamination in the region [76]. These inexpensive methods of drying are unhygienic, as they are characterised by insufficient maize qualities due to exposure of maize to rain and rewetting (which increases the moisture content of the grains), dust, foreign objects and insects, which may also serve as sources of the mould inoculum, whereas poor handling leads to non-uniform drying of the grains. Before storage, proper drying techniques need to be applied to reduce post-harvest losses and mycotoxin contamination. Rapid drying of crops to a low moisture level below 15% within 24–48 h will result in little or no fungal growth, and the products can be stored for longer periods of time [77,78,79]. Awuah and Ellis [80] in their studies show the importance of low moisture content in groundnuts at storage irrespective of whatever treatment method was applied. Their study revealed that 6.6% moisture level in groundnuts rendered them fungi-free for 6 months, regardless of the local storage protectant used, while at 12% moisture, only storage bags with the plant *Syzigum aromaticum* as a protectant effectively suppressed the fungi. Upon increasing the moisture content to 18.5%, treatment with *Syzigum aromaticum* was not as effective. Turner, Sylla [79] found that thorough drying and proper storage of agricultural produce in subsistence farm settings in West Africa can achieve 60% reduction in mean levels of mycotoxins. It is worth noting that in as much as drying is crucial, drying methods influence mycotoxin contamination levels; for example, above-ground drying of harvested produce reduces mycotoxin infection levels with fungal spores from the soil, while both drying in ventilated structures in the field and on plastic sheets reduce mycotoxin infection levels [81,82].

Sorting techniques are able to reduce the contamination of mycotoxin from 80% to 40% [83]. Handpicking or automatic sorting of discoloured, or mouldy kernels, winnowing, washing, and crushing, combined with de-hulling of maize grains, are effective in achieving significant removal mycotoxins of before storage [57,69,82,84]. Sorting can be carried out while in the field, during drying, and or when the grains are in storage. In addition, flotations of kernels reduce mycotoxin contamination by as much as 95% [85]. 

Microbial fermentation processes have also been reported to reduce mycotoxins in maize-fermented products [86,87]. Other advanced mycotoxin reduction techniques that have been applied in some parts of Africa include ozonation of peanut in a fumigation chamber, where about 70% reduction in mycotoxins in peanut was recorded [88]. 

Depending on the level and type of mycotoxin contamination of the food, some post-harvest processing methods are insufficient to eliminate aflatoxins from contaminated food but will definitely lead to the reduction or degradation of mycotoxins during processing [20,89,90,91]. In Africa, manual harvesting with non-removal of the husk is common among small farm holders, which is a good agricultural practice. In addition, handpicking and manual sorting of discoloured, mouldy, and foreign particles in the field, during drying and storage, are common practices and have proved to be very effective in the reduction in mycotoxin contamination. Fermentation of food crops is a common practice in Africa, and there are several fermentation products with different local names, which differ from one country to another.

#### 2.2.3. Storage Techniques

Crop loss is estimated to be between 20% and 30% in sub-Saharan Africa during storage [92,93]. Crop quality depends on storage; thus, poor storage causes contamination [94,95]. In sub-Saharan Africa, farmers continue to use traditional or crude storage methods that increase losses and mycotoxin contamination [69,96,97]. Cost of acquisition of storage facilities is one of the most important deciding criteria for farmers that has enabled them to hold on to these traditional methods [98]. These traditional storage facilities include: jerry-cans, where maize can still be damaged by pests; closed containers, which can encourage the growth of moulds and mycotoxin production; sacks, which are also highly attacked by rats and other pests; fragile pots, which can easily break; wooden cribs, which can be easily attacked by insects; and granaries [99]. In addition to these traditional storage facilities, grain protectants, such as *Syzigum aromaticum* plant and dried neem leaf powder with antifungal activities, among others, are added during storage [100]. Stored unshelled peanuts were found with reduced levels of aflatoxin contamination compared with stored shelled peanuts [101]. 

The metal silo appears attractive and has very positive attributes due to its durability; however, these positives may not outweigh their high initial cost since it is difficult for small farm holders to acquire one. Maize stored in purdue improved crop storage PICS bags (in clean stores) offers the same advantages as the metal silo, as follows: they cannot be penetrated by rats; they offer a lower initial cost; and they are most promising for solving storage issues in Africa. Walker, Jaime [102] state that proper moisture management is critical in controlling aflatoxin contamination since most farmers in Africa use crude methods such as teeth testing to estimate moisture. Periodically checking the temperature, as well as performing visual checks of stored maize, is recommended for evidence of fungal growth and to allow separation of the infested or infected portion. Cleanliness to prevent insect infestation and disease infection should always be maintained. The first-in first-out principle should be followed during storage and use of maize. Maize should be packed in clean appropriate hermetic or PICS bags or in sealed containers to avoid exposure to excessive moisture and humidity. 

### 2.3. Mycotoxin Detection Techniques

Mycotoxin toxicity occurs at very low concentrations. There is need, therefore, for sensitive and reliable methods for its detection at such low concentrations. Once the mycotoxin concentrations are known, chances of the population consuming highly contaminated food can be reduced and strategies for reducing the levels can be put in place. One cannot manage what one cannot measure; hence mycotoxin assessment is essential for a mycotoxin management strategy. Mycotoxin mitigation will not be much of an issue in Africa if there is accurate data from different regions to help make informed future decisions. Most African countries do not have data on mycotoxins, while several countries that initiated the acquisition of such data could not get much of the required information due to one or more of the following reasons: inadequate research funds; lack of advanced laboratory facilities; lack of capacity and expertise in the field of toxins; corruption; limited surveillance systems; and inability to prioritise mycotoxins in most African countries. For example, research has been conducted in recent decades on mycotoxin contamination in food in Botswana, Lesotho, and Swaziland [103,104,105]; however, there are no recent data, making follow-up and mitigation processes difficult. As Robens and Cardwell [106] clearly put it, the cost of analytical services to monitor levels of mycotoxin in both contaminated and uncontaminated crops is a substantial part of the overall cost of mycotoxin contamination. Acquisition of accurate data will imply using state-of-the-art mycotoxin analytic techniques, which, for the most part, are sophisticated, and it is a challenge as these instruments are rarely available, and when available, it is difficult to maintain them. Thus, developing the technical expertise of African nationals on maintenance and management of these sensitive analytical instruments will be a pro-active move. 

The importance of analysing techniques cannot be over emphasised. Before the long-held notion that 25% of the world’s crop is affected by mycotoxins by FAO (2004b), Miller [5] found that due to mycotoxin contamination, 40% of crop productivity was lost to diseases in developing countries. Recently, improvements in monitoring techniques have painted a different picture altogether, with higher figures (60–80%) of food crops lost due to mycotoxin contamination globally [4]. There is a need for efficient, cost-effective screening and analytical methods that can be used for the detection of mycotoxins in developing countries. There are several methods used to detect mycotoxins in food samples and biological samples. 

Chromatographic separations, coupled with a suitable detection system, are known conventional methods used to detect mycotoxins [107,108]. Chromatography associated with mass spectrometry becomes more effective as it allows lots of information to be obtained about the analyte, making its identification certain [109]. Depending on the purpose and feasibility of a method, different techniques (screening and or analytical) have been applied in Africa, such as the high-performance liquid chromatography (HPLC), liquid chromatography mass spectrometry mass tandem (LC-MS/MS), thin layer chromatography (TLC), fluorometry, immunochromatographic assay, enzyme-linked immunosorbent assay (ELISA), and lateral flow immunoassay (LFIA), among others (Table 2). Commonly used techniques for the determination of mycotoxin exposure are high-performance liquid chromatography with fluorescence detection (HPLC-FLUO) and high-performance liquid chromatography, including sequential mass spectrometry (LC-MS/MS) [110]. The advantage of LC–MS/MS is the ability to quantify trace level contaminants in food and feed other parent compounds and their metabolites [111], while HPLC is vastly used for chromatographic fingerprinting of the constituents, where mycotoxin standards are used in solvent calibration, thereby preprograming the HPLC for the targeted mycotoxin. The chromatographic techniques are very expensive, time-consuming, and require a high degree of expertise. Methods such as ELISA, fluorometry, and TLC have become routinely used tools for rapid monitoring of most mycotoxins. The advantages of these methods are speed, ease of operation, not requiring experts, sensitivity, and availability of test kits for most of the major mycotoxins used in Africa. 

Choosing an analytical technique is key in obtaining accurate data. Most African countries lack sophisticated techniques, such as high-performance liquid chromatography; however, preliminary testing techniques, such as thin-layer chromatography or dipsticks, have been used for screening; thus, positive samples could be followed up with a confirmatory high-performance liquid chromatography analysis, as recommended by Mwanza, Abdel-Hadi [112]. In all, the correct or actual incidence of occurrence of mycotoxins in a sample is a function of the sensitivity of the analytical method used. The cost of analytical services to monitor levels of mycotoxins in both contaminated and uncontaminated crops is a considerable part of the total cost of mycotoxin contamination. Sensitive, quantitative, and reliable analytical methods, whose uses were precluded in many developing countries because of cost, are now commonly used in several African countries (Table 2). Some African countries have applied both screening and analytical techniques for the detection and quantification of mycotoxins; such data can be reliable, unlike data from only screening methods. Independent reports of mycotoxin contamination in foods from different methods and individuals do corroborate, making the data reliable and confirming that most of the researchers have the required expertise.

**Table 2 toxins-14-00318-t002:** Techniques that have been used to detect mycotoxins in African countries.

Country	Commodity	Analytical Method	References
Angola	Maize	HPLC	[113]
Burkina Faso	Maize	LCMS/MS	[114]
	Maize	HPLC	[115]
	Infant cereal formula	HPLC	[116]
Cameroon	Feed	LC-ESI-MS/MS	[117]
	Feed	Fluorometry	[118]
	Maize products	ELISA	[119]
Côte d’Ivoire	Maize	HPLC	[120]
	Maize	LC-ESI-MS/MS	[121]
	Maize	UHPLC-MS/MS	[122]
Egypt	Feed	HPLC	[123]
	Cereal	TLC	[124]
	Maize	HPLC/TLC	[125]
Ghana	Maize	TLC	[126]
	Maize	Immunoassay	[127]
	Maize	HPLC	[128]
Kenya	Maize	LCMS	[129]
	Maize	ELISA	[130]
	Feed	HPLC	[123]
	Maize & its products	TLC	[131]
	Feed	TLC	[131]
	Maize	HPLC	[132]
Lesotho	Maize	HPLC	[133]
Malawi	Maize	Immunochromatographic assay	[134]
	Maize	LCMS/MS and HPLC	[38]
	Maize based beer	LCMS/MS	[135]
Mozambique	Maize	LCMS/MS	[114]
Nigeria	Maize	LCMS/MS	[136,137]
	Maize	LCMS/MS	[138]
Rwanda	Maize	Reveal Q+ and Accuscan Gold Reader	[139]
	Maize	ELISA	[140]
	Feed	ELISA	[140]
South Africa	Commercial maize	HPLC, LCMS/MS	[46]
	Feed	LCMS/MS	[44]
	Feed	HPLC/TLC	[45]
	Commercial maize	LCMS/MS	[47]
	Feed	LCMS/MS	[141]
Sudan	Feed	HPLC	[123]
	Feed	HPLC	[142]
Tanzania	Maize	HPLC	[143]
	Maize	UHPLC-MS/MS	[144]
	Maize	LCMS/MS/ELISA	[145]
Togo	Maize	HPLC-MS/MS	[146]
	Maize	Fluorometry	[147]
Tunisia	Cereal	HPLC	[148]
	Cereal	ELISA	[149]
Uganda	Maize	Fluorometry	[150]
	Maize	Fluorometry/TLC	[151]
	Maize	TLC	[152]
Zambia	Maize	Immunochromatographic assay	[19]
	Maize	ELISA	[153]
Zimbabwe	Maize	HPLC	[154]
	Maize	LCMS/MS	[155]

### 2.4. Mycotoxin Regulatory Limits

Mycotoxin is a global food safety problem, as recognised by the World Health Organisation [156], with subsistence farming communities being the populations most at risk of exposure. Most African countries lack their own regulatory limits, and this is attributed to insufficient scientific data (occurrence, exposure, and toxicological), and thus they depend on developed economies such as the European Union, the United States of America, and the Codex Alimentarius Commission. The disturbing issue is whether these regulatory standards borrowed from the West are protective enough! There is no certainty with regard to these standards; however, what is certain is the fact that most of these standards do not take into consideration the consumption habits of locals; hence, they do not reflect local reality. The non-proactiveness of food safety bodies (only 15 African countries have mycotoxin regulations [157]) and the complex agricultural farming system further complicate the mycotoxin situation in Africa. Since the 2003 FAO report of 15 countries in Africa having regulatory limits, much has not changed. Matumba, Sulyok [38] reported rare or non-existent regulatory standards; even when such standards exist, the capacity to enforce them is always lacking. The little change observed in regulatory standards since the 2003 FAO report are, for example, countries that had regulatory standards for food only have improved to include regulatory standards for dairy, food, and or feed. For example, Kenya and Nigeria have added regulatory standards for dairy, while Senegal, whose regulatory limits were developed for feed only, now include food. Much has really not changed in this regard, as reinforcement of regulatory standards in African countries is not be an easy task for food security reasons [158]. When setting up regulatory limits per region or country, multiple co-exposure in food should not be ignored [159]. Furthermore, undetected conjugated forms (masked) of mycotoxins that can hydrolyse into free toxins in the digestive tract should also be considered [160,161] when setting these regulatory limits. Although the metabolic fate of modified mycotoxins still remains a matter of scientific discourse, these undetected conjugated mycotoxins, also referred to as masked or modified mycotoxins, may be: matrix-associated; biologically modified by plants, animals, or fungi; or chemically modified by thermal or non-thermal processing [162]. Aflatoxins are the most regulated mycotoxins in Africa; other regulated mycotoxins include the following: fumonisin; patulin; ochratoxin A; deoxynivalenol; and zearalenone. Table 3 shows African countries with mycotoxins regulations. Most countries in Africa do not have mycotoxin regulatory standards and most often borrow from other countries and/or organisations solely for trade purposes. Regulatory limits of the United States of America and the European Union have been added to Table 3, as they are the main sources these regulatory limits are borrowed from.

Regulation on its own might not resolve this issue at the household level, as it would likely have little impact on subsistence-based farming in Africa. Ayalew, Hoffmann [174] state that regulations are only enforced in African countries for crops destined for export markets, although most African countries are primarily focused more on domestic and regional markets than on exports [175]. Ambler, De Brauw [176] found that farmers who reported on the quality of their crops (loss of quality) predominantly did not dispose of them, but diverted them from sale and seed for personal consumption, due to food security reasons. Niyibituronsa, Mukantwali [69] revealed that 96% of buyers do not respect mycotoxin safety standards; only 4% have some knowledge about these standards. Information, such as consumption habits, degree of diversification of diets, and common processing techniques, among others, are needed when setting mycotoxin regulatory limits per region and or within countries or within trading countries. There is a need for critical evaluation of interventions, taking into consideration, sustainability, cultural acceptability, economic feasibility, and ethical implications of all actors along the food production chain. This will help develop regional mycotoxin regulatory limits instead of using those of developed countries.

### 2.5. Socio-Political Factors

The socio-economic status of most inhabitants in sub-Saharan African countries predisposes them to the consumption of mycotoxin contaminated products, either directly or at various points in the food chain. The chronic health risks of mycotoxins are prevalent in Africa since mycotoxins occur more frequently under tropical conditions, and staple diets in the region are often susceptible to these substances [38,135,177]. The threat is intensified by the fact that staple diets in many African households are based on cereal crops, such as maize, which are highly susceptible to mycotoxin contamination; inability to diversify diets is one of the reasons for common health consequences. Most often, primary staples are also the main cash crops; produce with the best quality is often exported, leaving poor quality crops for home consumption, brewing of beer and sale in the informal sector [38,135], predisposing the population to consumption of mycotoxin contaminated foods [12].

Another possible route of exposure to mycotoxins in Africa is through trade. A Biomin survey on the global mycotoxin threat revealed high incidences in most samples from Africa [58]. Fungi can easily spread from one area to another, and considering that there are no strict regulations and control systems on mycotoxins in the region, sub-Saharan Africa, most often, is exposed to contaminated foods and products through global trade. The absence of strict regulations and control systems in African countries can be traced to the 1970s, considering the fact that the economies of these developing countries were highly distorted because of excessive government intervention and control. These countries experienced serious economic difficulties, with high inflation, unmanageable balance of payments and fiscal deficits, high external debt ratios, and negative Gross Domestic Product growth rates that were unable to match increases in population numbers. In the course of addressing these problems by the International Monetary Fund, the World Bank and other institutions initiated the structural adjustment programmes (SAPs) to provide balance of payments support to developing countries on the condition that they adopt this reform. The SAPs resulted in liberalised trade and exchange rate regimes and radically reduced subsidies in many developing countries. While trade liberalisation has no doubt yielded the targeted result (which was to boost economies), the effects of mycotoxins within these regions should not be neglected. In an attempt to resolve the issue at the time, another huge problem was created unknowingly, as very little was heard of the effects of climate change in the seventies. Mycotoxin regulatory limits are barriers to free trade, especially in countries where contamination levels are high, such as in Malawi, where the government prefers to export to countries with less strict or no mycotoxin legal limits [178].

The sudden onset of emergencies or instability, such as natural disasters, wars, and other political challenges, allows the collapse of food production and distribution systems, leading to food shortages. Africa is one of the regions in the world where there is frequent occurrence of wars, political conflicts, and natural disasters. Food aid is often an essential component of humanitarian response in such emergencies, where the African country concerned ends up getting food donations. In such situations, food aid or donations, especially sourced from donor countries, where mycotoxin contamination of the food is not assessed, could predispose the population to mycotoxin contamination.

Factors that are fundamental to a country’s ability to protect its population from mycotoxins include the political will to address exposure through the right institutions and the capability to test food for contamination. There is need to involve institutions—such as ministries of agriculture, research and scientific development, trade and industry, health, and other stakeholders directly or indirectly linked to the food production chain—in the decision-making process of mycotoxin mitigation in a country or within a region. There is also the need for countries within a region to build networks of trading relationships and harmonisation of policies and regulations to support any proposed interventions on regulatory limits [179,180]. Furthermore, since it can be expensive for a country to own and run its own research or analytical laboratory, there is need to create regional reference testing laboratories through stakeholders’ advocacy and regional partnerships within countries. This will ensure a coordinated response approach, while postgraduate training using state-of-the-art infrastructure will ensure sustainability. Growing the interest of the African scientific community towards increasing the research output in the country or region is imperative. Hence, scholarships should be granted to conduct research in mycotoxin-related fields. If most African countries still find it difficult to reinforce regulatory limits for the different mycotoxins, this could be due to the free trade within the region that they enjoyed, and it is high time for the government, in partnership with mycotoxin research organisations/individuals, to intervene and create awareness in its population of the socio-economic consequences of consuming mycotoxins. In most African countries, nepotism and tribalism, among others, are common challenges faced by Africans. Political appointees to head institutions of research involving mycotoxins are often flawed. These research institutions or regulatory bodies end up being headed by unqualified individuals, leading to their collapse.

### 2.6. Awareness

Despite tremendous efforts made to reduce the incidence of these natural toxins in foods and feeds since their discovery in the 1960s, they are still widely distributed at high levels and raise serious public health/economic concerns, especially in Africa. For an efficient mitigation of the risk posed by these mycotoxins to humans and animals, especially in Africa, it is critical for all stakeholders along the food supply chain to understand the health and economic risks of the presence of mycotoxins in their supply of food—anything short of that will just be yet another cosmetic solution. The effects of mycotoxins in food are unknown by many Africans. For example, Suleiman R. A. [67] evaluated the degree of awareness of mycotoxin contamination of food among some Tanzanian stakeholders and found out that over 80% of farmers did not have any knowledge or had never heard about mycotoxin contamination before and that about 67% of traders had no knowledge of mycotoxins contamination, while most consumers had no knowledge. Niyibituronsa, Mukantwali [69] conducted similar studies in Ethiopia and found low awareness of mycotoxins among stakeholders; about 60% of respondents were not aware of aflatoxins, while 40% had heard about it. Among the 40% who were aware of aflatoxins, about 37% knew its source, while less than 1% of respondents knew the effects. In a survey conducted in some districts in South Africa, 90% of respondents were not aware of mycotoxins and their consequences to humans and animals [98]. Another survey in Uganda on aflatoxin in groundnuts revealed that despite the fact that 61% of households indicated that they had indeed heard of aflatoxin, 75% of households dried groundnuts on the open earth at home, with only 3% using a tarpaulin and 10% using a pavement [181]. Other authors have reported similar findings. For example, in Malawi, Matumba, Monjerezi [10] conducted a study in the region with the highest rate of mycotoxin contamination in crops. The study revealed that 88% of the population understood that moulds posed a risk to human health and that few understood the risk, while half believed any toxins would be destroyed by basic cooking, which is not always the case, as food processing techniques have proved insufficient for eliminating some mycotoxins from contaminated food and feed due to their heat-resistant nature [20,89]. On the contrary, other researchers found that mycotoxins can be eliminated by the cooking process [182]. There is discrepancy on this issue; however, there should be a contamination level at which minor processing techniques would be able to eliminate mycotoxins depending on available data of the country, such as type of mycotoxin and the concentration levels in food. Education remains the key at this point in time as the authors of [67] found that level of education seems to be directly related to awareness of mycotoxins, as people who are more educated are more aware of the risks associated with food safety.

If mycotoxin contamination of food is a serious concern in Africa, governments and or research institutions in the Continent are not fully ignorant of its consequences but have chosen to prioritise other issues affecting the continent that need immediate attention, such as food insecurity, civil wars, education, health, corruption, gender inequality, child abuse, unemployment, and crime, among others. Due to corruption in most African countries, unqualified individuals with little background knowledge on food toxins are appointed to lead these institutions, resulting in weak regulatory bodies. It is rather unfortunate that leading authors in the field of mycotoxin research do not reflect the main regions where mycotoxin contamination is an issue for people’s health, such as in Africa [183]. As mentioned earlier, there is paucity of research data on mycotoxins in most African countries due to the limited number of studies conducted on dietary mycotoxins in the region compared with other regions of the world. This can be attributed to a lack of advanced laboratory equipment; inadequate research funds, capacity and expertise; and limited surveillance systems, thus making mitigation efforts a challenge. 

Mycotoxin contamination of food is a serious public health issue that endangers lives—for example, the loss of 125 lives in Kenya in 2004 and 16 lives in Tanzania 2016 as a result of consumption of AF-contaminated foods [184]. Liver cancer or other negative health effects related to food consumption and mycotoxin as highlighted by Omotayo [185] should be highly publicised in primary healthcare units in Africa [38,180]. It is important to differentiate the urban from the rural consumer. Thus, there is need to change the behaviour and consumption patterns of rural households. Furthermore, it is imperative for food safety purposes for all stakeholders along the food supply chain to understand in detail the health and economic risks associated with exposure to filamentous fungi and their toxins [186,187]. Awareness by farmers will improve their knowledge on mycotoxin control, such as the selection of suitable varieties, improving post-harvest management (sorting, drying, and the monitoring and control of moisture and rapid detection), and good storage and distribution practices. Awareness by traders will ensure that set regulatory standards are respected.

## 3. Conclusions

Mycotoxin contamination is a public health concern that endangers human/animal health. Apart from cancers, other common diseases have been associated with mycotoxin exposure, such as kwashiorkor, stunting, pulmonary fibrosis, hepatitis B and C, and even HIV, among others. With the growing concern of food insecurity on the continent, most Africans view the mycotoxin contamination food problem from the perspective of wasted food due to mould infestation rather than the health hazards associated with it. Either way, there are food insecurity problems based on safety and adequacy. This review provided a detailed explanation of six main factors whose direct or indirect contribution has led to high levels of mycotoxins in African foods. Ultimately, climate change remains the primary factor for high levels of mycotoxin in African foods. Other factors are partly man-made and can be manipulated to become a more profitable or less climate-sensitive response. In addition to climate change, farming systems, pre-harvest and post-harvest processing, and storage techniques, to an extent, can be considered primary contributing factors to high mycotoxins contamination in food; there is a thin line between these two main factors. Secondary contributing factors to high levels of mycotoxins in food include but are not limited to regulatory limits, mycotoxin detection techniques, socio-political factors, and awareness. Appropriate mastery and management of these factors is what will make a difference on the degree of mycotoxin contamination in food from one region of the world to another. 

Climate change is real. If it was predicted that aridity in Africa will increase between 5% and 15% in the 21st century—so too should be, for example, improvement in farming systems and or techniques, such as irrigation or supplemental irrigation and the use of insecticides to minimise the effects of this aridity on net productivity and diseases. In order to mitigate these changes, there is need for in-depth research on these toxins to provide accurate data in order to create awareness of the type of mycotoxins common in a given geographic location or country—creating awareness of the health and economic implications of consuming mycotoxin contaminated foods. Acquisition of accurate data will imply using state-of-the-art mycotoxin analytical techniques, as mycotoxin toxicity occurs at very low concentration levels and constitutes a substantial part of the overall cost of mycotoxin contamination research. Most African countries have started acquiring sophisticated equipment; however, much more still has to be carried out in this regard, such as training researchers in the field from the region. It is only after accurate data of a given geographic location have been acquired that proper regulatory limits can be put in place to reflect the local reality. There is a need for African leaders to put aside nepotism and tribalism when appointing individuals to head research institutions of such magnitude. Collaboration within regional laboratories and organising research workshops will go a long way in reducing mycotoxin contamination of food. 

## 4. Research Methodology

A literature review was conducted, using the PRISMA (Preferred Reporting Items for Systematic Reviews and Meta-Analyses) guidelines [188] to gather information on mycotoxin-related issues in Africa. Key words/phrases used to access information were: mycotoxin; Africa; sub-Saharan Africa; climate change; farming systems; regulatory limits; food; socio-political; trade; awareness; legislation; and detection techniques. A comprehensive literature search was performed using Google Scholar to extract peer-reviewed studies on mycotoxin-related issues, specifically in Africa, published between 2005 and 2021. Very few studies on mycotoxins had been conducted in Africa beyond this period, and most of the research had been acknowledged in later studies. Articles, conference papers, and book chapters published in English, with relevant information, were downloaded and analysed. Out of the 142 articles downloaded, 57 did not have the required information. Articles not written in English were discarded, while 85 were considered for this review.

## Figures and Tables

**Table 1 toxins-14-00318-t001:** Effects of climate change on maize production in South Africa (2005–2020).

Climate Change in South Africa			
Year	Production of Maize/Tons	Annual Rainfall/mm	Temperature Change/°C
2005	11,715,948	395	0.9604
2006	6,935,056	566	0.5105
2007	7,125,000	424	0.7655
2008	12,700,000	437	0.8384
2009	12,050,000	472	0.7332
2010	12,815,000	474	1.2107
2011	10,360,000	540	0.5503
2012	12,120,656	462	0.6957
2013	11,810,600	420	0.7135
2014	14,250,000	449	0.9467
2015	9,955,000	368	1.5954
2016	7,778,500	423	1.6038
2017	16,820,000	424	1.0127
2018	12,510,000	383	1.1953
2019	11,275,500	382	1.7086
2020	-	460	0.9330

**Table 3 toxins-14-00318-t003:** Regulatory limits of mycotoxins in Africa, the European Union, and the United States of America.

Country	Commodities	Mycotoxin	Maximum Acceptable Level	References
Algeria	Peanuts, nuts, and cereals	Aflatoxin B1	300 µg/kg	[163,164]
	Cattle feed	Aflatoxin B1,G1,B2,G2	20 µg/kg	
Côte d’lvoire	Straight feedstuffs	Aflatoxin B1,G1,B2,G2	100 µg/kg	[163,164]
	Complete feedstuffs	Aflatoxin B1,G1,B2,G2	10 µg/kg	
	Complete feedstuffs for pigs/poultry	Aflatoxin B1,G1,B2,G2	38 µg/kg	
	Complete feedstuffs for cattle/sheep/goats	Aflatoxin B1,G1,B2,G2	75 µg/kg	
	Complete feedstuffs for dairy cattle	Aflatoxin B1,G1,B2,G2	50 µg/kg	
Egypt	Peanuts, oil seeds, and cereals	Aflatoxin B1	5 µg/kg	[163,164]
		Aflatoxin B1,G1,B2,G2	10 µg/kg	
	Corn	Aflatoxin B1	10 µg/kg	
		Aflatoxin B1,G1,B2,G2	20 µg/kg	
	Animal and poultry fodder	Aflatoxin B1	10 µg/kg	
		Aflatoxin B1,G1,B2,G2	20 µg/kg	
Kenya	All foods	Aflatoxin B1	5 µg/kg	[9,37,163,165,166]
	Milk and milk products	Aflatoxin M1	0.05 µg/kg	
	Peanuts, products, and vegetable oils	Aflatoxin B1,G1,B2,G2	20 µg/kg	
	All foods	Fumonisins	2000 µg/kg	
Malawi	All foods	Aflatoxin B1,G1,B2,G2	10 µg/kg	[164,165]
	All foods	Aflatoxin B1	5 µg/kg	
	Peanuts for export	Aflatoxin B1	5 µg/kg	
	All foods	Fumonisins	2000 µg/kg	
Mauritius	All foods	Aflatoxin B1	5 µg/kg	[163,164]
		Aflatoxin B1,G1,B2,G2	10 µg/kg	
	Groundnuts	Aflatoxin B1	5 µg/kg	
		Aflatoxin B1,G1,B2,G2	15 µg/kg	
Mozambique	Peanuts, peanut milk	Aflatoxin B1,G1,B2,G2	10 µg/kg	[164]
	Peanuts, maize, peanut butter, cereals and feedstuffs	Aflatoxin B1,G1,B2,G2	10 µg/kg	
Nigeria	Cereals and cereal products	Aflatoxin B1,G1,B2,G2	4 µg/kg	[164,167]
		Aflatoxin B1	2 µg/kg	
	Feedstuffs	Aflatoxin B1	50 µg/kg	
	Nuts, peanuts, and almonds	Aflatoxin B1,G1, B2, G2	4–5 µg/kg	
	Peanut products as straight feedstuffs	Aflatoxin B1,G1,B2,G2	50 µg/kg	
	Melon	Aflatoxin B1	2 µg/kg	
		Aflatoxin B1,G1,B2,G2	4 µg/kg	
	Infant foods	Aflatoxin B1,G1,B2,G2	1–2 µg/kg	
	Fluid milk and its products	Aflatoxin M1	0.5 µg/kg	
	Unprocessed cereals	Fumonisins	<1000 µg/kg	
	Unprocessed maize		1000 µg/kg	
	Maize for human consumption		4000 µg/kg	
	Raw cereals	Ochratoxin A	0.5 µg/kg	
	Wine and juice		2 µg/kg	
	Unprocessed cereals		5 µg/kg	
	Spices		20 µg/kg	
	Processed cereal-based foods	Deoxynivalenol	200 µg/kg	
	Cereal grains		2000 µg/kg	
	Unprocessed cereals		1750 µg/kg	
	Unprocessed cereals for human consumption	Zearalenone	100 µg/kg	
	Unprocessed maize		350 µg/kg	
	Cereals intended for human consumption		75 µg/kg	
Senegal	Peanut products as feedstuff	Aflatoxin B1	300 µg/kg	[163]
	All foods	Aflatoxin B1,G1,B2,G2	20 µg/kg	
South Africa	All foods	Aflatoxin B1,G1,B2,G2	10 µg/kg	[164,168]
	All foods	Aflatoxin B1	5 µg/kg	
	Milk and milk products	Aflatoxin M1	0.05 µg/kg	
	Corns and corn products	Fumonisin	100–200 µg/kg	
	All foods	Zearalenone	3000–5000 µg/kg	
		Patulin	50 µg/kg	
		Deoxynivalenol	2000 µg/kg	
		Fumonisin B1 & B2	4000 µg/kg	
Sudan	Oil seeds	Aflatoxin B1,G1,B2,G2	10 µg/kg	[164,165]
	Wheat	Ochratoxin A	15 µg/kg	
	All foods	Fumonisin	2000 µg/kg	
Tanzania	Cereals, oil seeds	Aflatoxin B1	5 µg/kg	[164,165,166]
		Aflatoxin B1,G1,B2,G2	10 µg/kg	
	Feeds	Aflatoxin B1	5 µg/kg	
		Aflatoxin B1,G1,B2,G2	10 µg/kg	
	All foods	Fumonisin	2000 µg/kg	
Tunisia	All products	Aflatoxin B1	2 µg/kg	[164]
		Aflatoxin B1,G1,B2,G2	4 µg/kg	
	Milk	Aflatoxin M1	0.05 µg/kg	
Zimbabwe	All foods	Aflatoxin B1	5 µg/kg	[163]
	All foods	Aflatoxin B1,G1,B2,G2	10 µg/kg	
	Poultry	Aflatoxin B1,G1	10 µg/kg	
European Union	Cereals and processed products, groundnuts, nuts, and dried fruits and processed products intended for direct human consumption	Aflatoxin B1	2 µg/kg	[166,169,170,171]
		Aflatoxin B1,G1,B2,G2	4 µg/kg	
	Groundnuts, maize to be subjected to sorting, or other physical treatment, before human consumption	Aflatoxin B1	8 µg/kg	
		Aflatoxin B1,G1,B2,G2	15 µg/kg	
	Milk and its products	Aflatoxin M1	0.05 µg/kg	
	Baby and infant food	Aflatoxin B1	1–2 µg/kg	
	Complete feedstuffs for cattle and sheep, excluding young ones	Aflatoxin B1	20 µg/kg	
	Complete feedstuffs for calves and lambs	Aflatoxin B1	50 µg/kg	
	Raw cereal grains (including raw rice and buckwheat)	Ochratoxin A	5 µg/kg	
	All products derived from cereals (including processed cereal products and cereal grains intended for direct human consumption)		3 µg/kg	
	Dried vine fruits (currants, raisins, and sultanas)		10 µg/kg	
	Fruit juices and fruit nectar, in particular, apple juice and fruit juice ingredients in other beverages	Patulin	50 µg/kg	
	Concentrated fruit juice after reconstitution as instructed by the manufacturer		50 µg/kg	
	Spirit drinks, cider, and other fermented drinks derived from apples or containing apple juice		50 µg/kg	
	Solid apple products, including apple compote, apple puree intended for direct consumption		25 µg/kg	
	Apple juice and solid apple products, including apple compote and apple puree, for infants and young children and labelled and sold as intended for infants and young children		10 µg/kg	
	Other baby food		10 µg/kg	
	Cereal products as consumed and other cereal products at retail stage	Deoxynivalenol	500 µg/kg	
	Flour used as raw material in food products		750 µg/kg	
	All feedstuffs containing unground cereals	Rye ergot	1,000,000 µg/kg	
	All foods	Fumonisins	1000 µg/kg	
United states of America	Feedstuff ingredients	Aflatoxin B1,G1,B2,G2	20 µg/kg	[166,172,173]
	Cottonseed meal intended for beef cattle swine and poultry feedstuffs	Aflatoxin B1,G1,B2,G2	300 µg/kg	
	Maize and peanut products for beef cattle swine or poultry	Aflatoxin B1,G1,B2,G2	100–300 µg/kg	
	Whole milk, low fat milk, and skim milk	Aflatoxin M1	0.5 µg/kg	
	Finished wheat products	Deoxynivalenol	1000 µg/kg	
	Grains and grain by-products		10 000 µg/kg	
	Grains and grain by-products for swine		5000 µg/kg	
	All foods	Fumonisins	2 000 µg/kg	
	Feed for horses		500 µg/kg	
	Feed for swine		10 000 µg/kg	
	Feed for beef cattle and poultry		50 000 µg/kg	

## Data Availability

This is a review paper and the data presented in this study are openly available in published papers listed in References.

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
