# Peer review of "Six Main Contributing Factors to High Levels of Mycotoxin Contamination in African Foods"

_toxins, 2022, doi:10.3390/toxins14050318_

Round 1
Reviewer 1 Report
This work makes a good revision of the most important factors contributing to high levels of mycotoxins in Africa. The subject is of high interest in terms of food safety and food security for African countries, and fits the scope of the journal. In general, the most important aspects are addressed, even if some could be more descriptive.
The revision is long, and this is mainly due to the repetition of ideas (and even of sentences) throughout the text (e.g. lines 197-199, 208-209, 221-223, 308-311, 322-329, 420-422, 600-604), more than because it addresses the issues with the expected depth.
References are not properly cited throughout the manuscript, and must be revised following the journal’s guidelines.
The English language is not always correct, and must be revised.
Some ideas seem contradictory, mostly in section 2.1 Climatic and environmental factors, maybe because different studies are cited with different results without a proper integrative discussion. E.g., lines 141-146 seem to have contradictory ideas, that must be justified.
Lines 153-154: the information made available does not support the conclusion presented in this sentence. The authors present the evolution of climatic conditions in South Africa throughout 15 years, but there is no long term information on mycotoxin contamination levels (this information is made available only for one year in this period of time). How come do the authors conclude that contamination is due to climate change based on one year of mycotoxin data?
In Table 1, the reported temperature change is compared to what, the previous year? Please clarify.
Lines 258-259: Is overcrowding considered a Good agricultural practice that should be avoided? Please explain or clarify.
Lines 265-266: “… appropriate application of insect pests…”. Do you mean “… appropriate control of insect pests…”?
Table 2: VICAM is not an analytical method by itself. In the Cameroon and Uganda examples, the method should be reported as fluorometry. Check all other examples where VICAM is cited. Why are there examples where no analytical method is cited (Togo and Cote d’Ivoire)? List countries by alphabetic order.
Table 3: References are not adequate. For example, the reference for the European Union is a paper from 1995. The CE regulation should be cited instead. The same for all others. Why are there countries for which there are no references (Mozambique, Sudan, Tanzania, Tunisia)? The cases of Sudan (see CODEX) and Tunisia (unknown) must be clarified. List countries by alphabetic order.
Section2.6 Awareness: References to mycotoxin health effects already established for African countries should be included in this section.
Section 3 Conclusion: the first sentence about health concerns should have been discussed (even if briefly) in the previous section.
Author Response
Point 1: What is the main aim of the section “2.3. Mycotoxin detection techniques”? According to mine opinion, it should be to represent an overview of the currently available methods for the mycotoxins determination in Africa. However, authors wrote several papers (references 118, 127, 129, etc) in which samples from Africa were analysed by method which are not developed and applied in Africa.
Response 1: A lot of mycotoxin data have been generated in Africa by African scientists and others by conventional and non-conventional methods. The aim of this section is to present these different methods that have been applied in Africa in order to make the decision wheather to trust the data generated by these scientists. In some cases sophisticated state-of-the-art mycotoxin analytic techniques have been applied, in others cheap, fast screen methods were used. Irrespective of where the method was developed, the idea was to discourse the pros and cons of the different methods and which ones are commonly used in Africa.
Point 2: Table 1: Please apply some adequate statistical evaluation to obtaine comparison/deifferences between compared values between years. Please reduce the number of digits for temperature, It is better to express production of maize in million tonnes, which is the most commonly used units for this.
Response 2: Temperature at certain years changes with the smallest digit possible, if not recorded to the smallest gradual change possible over time, sudden changes in value in the long long might become unexplainable.
Point 3: Some references are not properly used. Please check others references .For example, reference 37 is related to aflatoxins occurence in one Europe countrie, Serbia, but authors used this reference to explaine situation in Zambia:“In Zambia, for example, higher mycotoxin concentration in groundnut was primarily a consequence of climatic extremes, such as serious drought and hot summer temperatures [Kos, Janić Hajnal [37].”
Response 3: In Zambia, for example, higher mycotoxin concentration in groundnut was primarily a consequence of climatic extremes, such as serious drought and hot summer temperatures [Kachapulula et al., 2017].”
Point 4: Some sentences should be checked/changed in terms of meaning/grammar. For example: „Aflatoxins are the most regulated mycotoxins in Africa and include the following: fumonisin; patulin; ochratoxin A; deoxynivalenol; and zearalenone.” This sentence is not correct, because aflatoxins are not including fumonisin; patulin; ochratoxin A; deoxynivalenol; and zearalenone, they are all different mycotoxins.
Response 4: The sentence has been corrected. ‘Aflatoxins are the most regulated mycotoxins in Africa, other regulated mycotoxins include the following: fumonisin; patulin; ochratoxin A; deoxynivalenol; and zearalenone’.
Point 5: Instead of singular "aflatoxin and fumonisin" please use plural, aflatoxins and fumonisins. When aflatoxins are written in the singular then it is usually emphasized to which aflatoxins it refers (aflatoxin B1, or others).
Response 5: The plural of the mycotoxins have been applied in the article
Thank you

Reviewer 2 Report
Dear Authors,
Your manuscript entitled Six main contributing factors to high levels of mycotoxin contamination in African foods deals with a topic that could be of interest for readership at regional scale.
Overall, I find that it is beneficial to have some information on mycotoxin contamination in African food. The authors bring valuable information by including climatic factors, farming system and processing techniques etc.
The review follow a systematic and logical sequence making easy to be read.
The introduction section reveal your objective (factor that contributes to mycotoxin contamination), there are a lot of information specific to the topic.
The ’’body of the paper’’ is well developed, the state of the art in literature is well presented but can be improved with more recent references; try to put figures and/or tables to present your synthesis or to show key data taken directly from the original papers.
I suggest that the authors develop Discussion section and go deeply in the analysis and comparation of the techniques, limits of detection etc, exploiting the potential mechanisms causing the observed conclusion.
Finally, in conclusion try to including a brief summary of your key finding, future directions, highlight of your hypothesis as well as improvements compared to already reported work.
Thanks,
Best regards
Author Response
Response to Reviewer 1 Comments
Point 1: The ’’body of the paper’’ is well developed, the state of the art in literature is well presented but can be improved with more recent references; try to put figures and/or tables to present your synthesis or to show key data taken directly from the original papers.
Response 1: Tables have been improved and references added
Point 2: I suggest that the authors develop Discussion section and go deeply in the analysis and comparation of the techniques, limits of detection etc, exploiting the potential mechanisms causing the observed conclusion.
Response 2: The aim of this section is to present different methods that have been applied in Africa in order to make the decision wheather to trust the data generated by African scientists. In some cases sophisticated state-of-the-art mycotoxin analytic techniques have been applied, in others cheap, fast screen methods were used. Irrespective of where the method was developed, the idea was to discourse the pros and cons of the different methods and which ones are commonly used in Africa.
Point 3: Finally, in conclusion try to including a brief summary of your key finding, future directions, highlight of your hypothesis as well as improvements compared to already reported work.
Response 3: Ultimately, climate change remains the primary factor for high levels of mycotoxin in African foods. Other factors are partly man-made and can be manipulated to become a more profitable or less climate sensitive response. In addition to climate change, farming systems, pre-harvest and post-harvest processing and storage techniques, to an extent, can be considered primary contributing factors to high mycotoxins contamination in food; there is a thin line between these two main factors. Secondary contributing factors to high levels of mycotoxins in food include but not limited to regulatory limits, mycotoxin detection techniques, socio-political factors and awareness.
Appropriate mastery and management of these factors is what will make a difference on the degree of mycotoxin contamination in food from one region of the world to another. There is need for African leaders to put aside nepotism and tribalism when appointing individuals to head research institutions of such magnitude. Collaboration within regional laboratories and organising research workshops will go a long way to reduce mycotoxin contamination of food.
Thank you

Reviewer 3 Report
The Manuscript „Six main contributing factors to high levels of mycotoxin contamination in African foods” represents a good review of unfortunately not very good situation related to the mycotoxins situation in Africa. Besides several minor mistakes, listed below, the main mistake of this manuscript is related to the absence of the review of results related to the occurrence of different mycotoxins in food originated from different African countries. Please add one or two tables which will present the data of different mycotoxins occurrence, in different food from different African countries, and write discussion about collected data. The new tables will be an added value to the manuscript.
What is the main aim of the section “2.3. Mycotoxin detection techniques”?
According to mine opinion, it should be to represent an overview of the currently available methods for the mycotoxins determination in Africa. However, authors wrote several papers (references 118, 127, 129, etc) in which samples from Africa were analysed by method which are not developed and applied in Africa.
Table 1: Please apply some adequate statistical evaluation to obtaine comparison/deifferences between compared values between years. Please reduce the number of digits for temperature, It is better to express production of maize in million tonnes, which is the most commonly used units for this.
Some references are not properly used. Please check others references .For example, reference 37 is related to aflatoxins occurence in one Europe countrie, Serbia, but authors used this reference to explaine situation in Zambia:“In Zambia, for example, higher mycotoxin concentration in groundnut was primarily a consequence of climatic extremes, such as serious drought and hot summer temperatures [Kos, Janić Hajnal [37].”
Some sentences should be checked/changed in terms of meaning/grammar. For example: „Aflatoxins are the most regulated mycotoxins in Africa and include the following: fumonisin; patulin; ochratoxin A; deoxynivalenol; and zearalenone.” This sentence is not correct, because aflatoxins are not including fumonisin; patulin; ochratoxin A; deoxynivalenol; and zearalenone, they are all different mycotoxins.
Instead of singular "aflatoxin and fumonisin" please use plural, aflatoxins and fumonisins. When aflatoxins are written in the singular then it is usually emphasized to which aflatoxins it refers (aflatoxin B1, or others).
Author Response
Response to Reviewer 1 Comments
Point 1: The revision is long, and this is mainly due to the repetition of ideas (and even of sentences) throughout the text (e.g. lines 197-199, 208-209, 221-223, 308-311, 322-329, 420-422, 600-604), more than because it addresses the issues with the expected depth.
Response 1: Some of the reasons of high mycotoxins in African foods are intertwined or inter-related, reasons for the repetition of some ideas. Nevertheless, some unnecessary repetition has been minimised.
Point 2: References are not properly cited throughout the manuscript, and must be revised following the journal’s guidelines.
Response 2: References have now been cited following the journal’s guidelines.
Point 3: The English language is not always correct, and must be revised.
Response 3: English language editing has been done on this article
Point 4: Some ideas seem contradictory, mostly in section 2.1 Climatic and environmental factors, maybe because different studies are cited with different results without a proper integrative discussion. E.g., lines 141-146 seem to have contradictory ideas, that must be justified.
Response 4: These sentences are illustrations to how climate variables do affect mycotoxin production. Like the case of Malawi, where maize harvested from regions with high temperatures and low rainfall were highly contaminated with mycotoxins. Then on a continental trend, lower incidences of mycotoxins have been observed in north and southern Africa as compared to those in east and west Africa. Relating it to climate variables, north and southern Africa have a drier climate compared to east and west Africa with high temperatures and humid climate. The Serbia example was to affirm the fact that other scientists elsewhere have similar observations and to help make the conclusion strong enough.
Point 5: Lines 153-154: the information made available does not support the conclusion presented in this sentence. The authors present the evolution of climatic conditions in South Africa throughout 15 years, but there is no long term information on mycotoxin contamination levels (this information is made available only for one year in this period of time). How come do the authors conclude that contamination is due to climate change based on one year of mycotoxin data?
Response 5: In as much as there is limited data on mycotoxin, emphases was placed on the one year because the authors were involve in the research work but their conclusion was not based on that year alone. Meyer et al., 2019 had done a long term analysis of mycotoxins within which the one year we laid emphases fell within those years of their multi-mycotoxin analyses. Coincidentally, there was a drastic climatic changes that same year in South Africa. Hence, our conclusion was based on several years of mycotoxin data why placing emphases on that particular year not only because we were actively involved in the mycotoxin study for that year, but also because of the drastic climate changes that occurred in that same year.
Point 6: In Table 1, the reported temperature change is compared to what, the previous year? Please clarify.
Response 6: Yes, it is compared to the previous year, showing a gradual rise in temperature rise
Point 7: Lines 258-259: Is overcrowding considered a Good agricultural practice that should be avoided? Please explain or clarify.
Response 7: ‘Agricultural practices’ such as overcrowding of plants, should be avoided as it may lead to humid and warm conditions and NOT ”Good agricultural practices”
Point 8: Lines 265-266: “… appropriate application of insect pests…”. Do you mean “… appropriate control of insect pests…”?
Response 8: ‘appropriate control of insect pests’
Point 9: Table 2: VICAM is not an analytical method by itself. In the Cameroon and Uganda examples, the method should be reported as fluorometry. Check all other examples where VICAM is cited. Why are there examples where no analytical method is cited (Togo and Cote d’Ivoire)? List countries by alphabetic order.
Response 9: All analytical methods have now been cited
Point 10: Table 3: References are not adequate. For example, the reference for the European Union is a paper from 1995. The CE regulation should be cited instead. The same for all others. Why are there countries for which there are no references (Mozambique, Sudan, Tanzania, Tunisia)? The cases of Sudan (see CODEX) and Tunisia (unknown) must be clarified. List countries by alphabetic order.
Response 10: Countries listed in alphabetical order and the references added
Point 11: Section 2.6 Awareness: References to mycotoxin health effects already established for African countries should be included in this section.
Response 11: It has been included
Point 12: Section 3 Conclusion: the first sentence about health concerns should have been discussed (even if briefly) in the previous section.
Response 12: Sorted too.
Thank you

Round 2
Reviewer 1 Report
The revision made by the authors is very superficial and did not adequately reply to the previous reviewer's comments. In fact, only the minor comments requiring small and direct adjustments were addressed.
Comments on Table 1 and climate data were not properly addressed and were not subject of a sounder discussion.
The mention to VICAM as a method has been corrected in Table 2, but not in the text.
References in Table 3 are still not appropriate. I cannot accept that all the legislation for European Union and USA is not properly cited. The authors still use Pittet (1995) as reference!! There are numerous EC Regulations (Regulation 1881/2006 and subsequent amendments; Directive 2022/32/EC, Recommendation 2006/576/EC) and USA legislation.
Lines 615-620: the reference to mycotoxin-related health concerns in Africa has been poorly addressed. The added text does not make sense and proper references are lacking.
Author Response
Good day
Thank you for your detailed recommendations, all the areas you mentioned of concerned have been fully attended to, especially Table 3.
Thank you

Round 3
Reviewer 1 Report
Please check reference numbering. New references have been included, but apparently the reference numbering has not been properly updated (e.g. ref. 166. I have not checked the others...).
Table 3 - Please check the AFB1 value for baby and infant food. I can't seem to find this value in the most updated CE regulations... Some mycotoxins important in african foods have been left out from those listed table 3 in the European legislagion (fumonisins, zearalenone). Why is that? I understand that the authors don't need to include the full list of food products and regulated mycotoxins, but FMs and ZEA are very important in african cereals... In fact, these mycotox are listed for many african countries in Table 3.
Author Response
Good day
All the required recommendations are implemented. Thank you
